# A deep learning approach to private data sharing of medical images using conditional generative adversarial networks (GANs)

Hanxi Sun[1], Jason Plawinski[2], Sajanth Subramaniam[2], Amir Jamaludin[3], Timor Kadir[4], Aimee Readie[2], Gregory Ligozio[2], David Ohlssen[2], Mark Baillie[2], Thibaud Coroller[2]*

1 Department of Statistics, Purdue University, West Lafayette, IN, United States of America, 2 Novartis Pharmaceutical Corporation, East Hanover, New Jersey, United States of America, 3 Oxford Big Data Institute, Oxford, United Kingdom, 4 Plexalis Ltd, Oxford, United Kingdom

☯ These authors contributed equally to this work.
* thibaud.coroller@novartis.com

**Data Availability Statement:** The real datasets (F2305, A2209) used for training and testing were obtained from completed, anonymized clinical

## Abstract

Clinical data sharing can facilitate data-driven scientific research, allowing a broader range of questions to be addressed and thereby leading to greater understanding and innovation. However, sharing biomedical data can put sensitive personal information at risk. This is usually addressed by data anonymization, which is a slow and expensive process. An alternative to anonymization is construction of a synthetic dataset that behaves similar to the real clinical data but preserves patient privacy. As part of a collaboration between Novartis and the Oxford Big Data Institute, a synthetic dataset was generated based on images from COSENTYX® (secukinumab) ankylosing spondylitis (AS) clinical studies. An auxiliary classifier Generative Adversarial Network (ac-GAN) was trained to generate synthetic magnetic resonance images (MRIs) of vertebral units (VUs), conditioned on the VU location (cervical, thoracic and lumbar). Here, we present a method for generating a synthetic dataset and conduct an in-depth analysis on its properties along three key metrics: image fidelity, sample diversity and dataset privacy.

## Introduction

In recent years, deep learning has become an indispensable tool for clinical image analysis, including ophthalmological [1] and pathological [2] medical images [3, 4]. By the end of 2020, PubMed [5] had a total of 9,497 entries for "deep learning + image," with more than half of them published in 2020. While the number of publications using such techniques has exploded, the number of publicly available datasets has not. For example, to enable breakthroughs in areas beyond opthamlogy and pathology, researchers require access to informative and labelled datasets representative of the scientific problem to be addressed. Moreover, to solve problems in the biomedical space, often large, pooled datasets are required that may span multiple institutions. Sharing data enables scientific progress but doing so can be challenging due to privacy [6], ethical [7], legal [8], and institutional challenges [9] associated with the

trials. Since both the clinical trial data and the images are anonymized these are no longer personal data and so no further EC/IRB approvals are required. Furthermore a synthetic version of each dataset (with similar sample size and distribution) will be shared to ensure no reidentification of actual patients is possible. Paper code and synthetic datasets can be found at https://github.com/tcoroller/pGAN. Novartis is committed to sharing with qualified external researchers' access to patient-level data and supporting clinical documents from eligible studies. These requests are reviewed and approved based on scientific merit. All data provided are anonymized to respect the privacy of patients who have participated in the trial in line with applicable laws and regulations. This trial data availability is according to the criteria and process described on www.clinicalstudydatarequest.com.

**Funding:** The study was sponsored by Novartis Pharma AG. Novartis personnel and academic advisors from Oxford Big Data Institute (BDI) designed the project.

**Competing interests:** I have read the journal's policy and HS, AJ, TK declared no competing interests. I have read the journal's policy and TC, MB, JP, SS, GL, AR, DO are employees of Novartis.

dataset. Finding a solution to this would facilitate collaboration across research groups, allowing the possibility to reproduce and build upon existing work. A lack of data availability impacts reproducibility that requires access to code and data to be addressed. For example, most method papers are not published with the corresponding research data, which would allow other scientists to validate and verify results.

While replication [10] of hypotheses is the key to scientific understanding in the biomedical field, it is typically hindered by the need to access and analyse multiple data sources. Despite efforts to share data [11], there is a lack of labelled datasets on which to build novel models. Using a risk-based approach to anonymizing data [12] is an option, a process to remove or reduce information that will in turn reduce the risk of an individual being identified, but it is a complex and laborious task. One part of the process is to ensure that all information that could directly identify an individual is removed. While this is difficult enough for tabular data, it becomes even more challenging for images due to the complexity of the meta information and the need to exhaustively check each slice of the image for pixel-burnt information (e.g., patient name).

Other parts of the anonymization process can reduce data utility, which has a knock-on effect on the quality of subsequent research. Open scientific collaboration comes at a tension with the federation and control of sensitive data. Data sharing contributes to scientific understanding in an open and collaborative spirit. However, sharing data could have consequences if it was not properly approved (non-anonymized datasets, lack of patient agreement) or if it is leveraged for unintended uses. This illustrates the need for other data-sharing methods (including for models) that would protect scientists from this dilemma.

One solution is federated learning [13] in which instead of sharing data, a model is shared with multiple researchers to be trained on their private datasets, rather than pooling datasets in a single location. This allows the researchers to efficiently utilize a trained model across restricted datasets but is a rather complex process to setup to ensure efficient model training and tackling of security concerns. Another solution could be to share a synthetic version of a sensitive dataset that preserves global properties (for example, their association to a specific clinical variable) while preserving patients' privacy. While this method cannot fully replace an actual dataset, this approximative version could help address important steps in this research process. For example, one could use such a dataset to prototype new methods (i.e., define a model architecture). Furthermore, if this approximation could be generated in a quick and effortless way, it could become a viable option for increasing the data quantities needed for current models. This provides opportunities for changing how data are shared and provides sufficient control for data custodians, while allowing for the pooling and integration of disparate data sources to answer complex questions.

This paper proposes a method that uses generative adversarial networks (GANs) [14] to create synthetic images while conserving its association with clinical variables as shown in **Fig 1**. A synthetic dataset should have three core properties—namely fidelity, diversity and privacy—before it can be externally shared as a substitute for the original dataset. It should reproduce realistic samples in comparison to the original dataset (fidelity), the samples should exhibit similar variations as the original dataset (diversity), and the original samples should not be retrievable from the synthetic dataset (privacy).

## Materials and methods

The privacy GAN (pGAN) model utilized anonymized images from completed and anonymized secukinumab clinical trials to develop new datasets. The base dataset included 108 patients from the MEASURE 1 AIN457F2305 (F2305) study [15], in which imaging was done

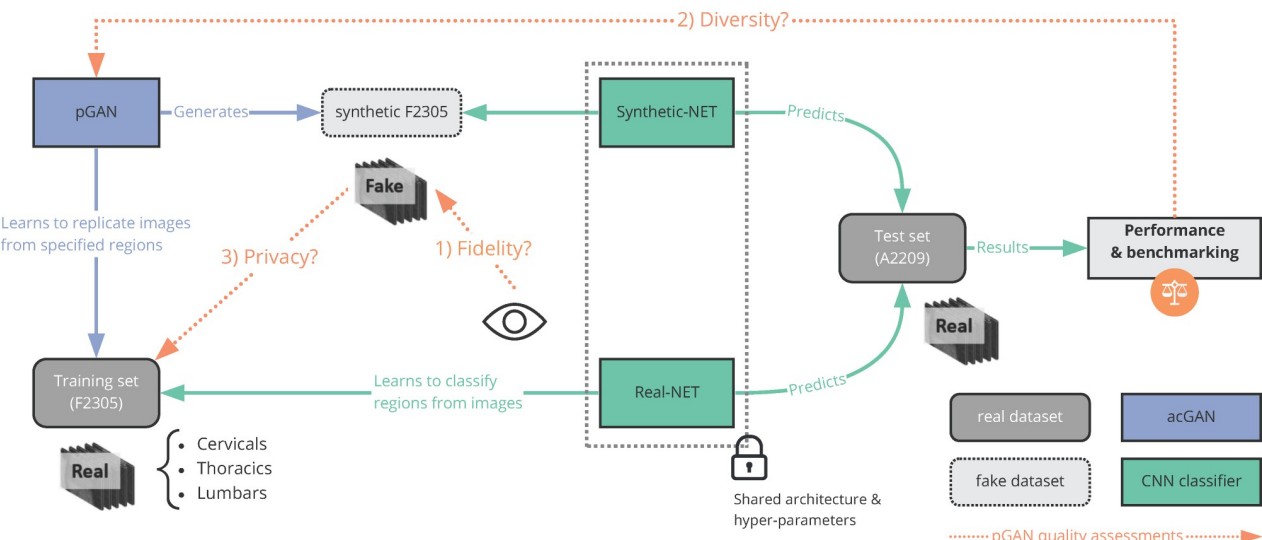

**Fig 1. Overview of the privacy GAN (pGAN) model workflow to share synthetic medical images (vertebral MRIs).** An auxiliary classifier generative adversarial network (ac-GAN) was trained from a dataset composed of vertebral images alongside their corresponding locations (cervical, thoracic, lumbar). Subsequently the pGAN model was trained to generate a synthetic dataset of vertebrae. The quality of the output synthetic dataset was evaluated based on three criteria (privacy, fidelity & diversity). Once all criteria passed, the synthetic data could safely be shared with external scientists without any privacy risk while retaining all data usefulness.

using T1 and STIR sagittal MRI at baseline and Weeks 16, 52, 104, 156 and 208. STIR images were not included in this analysis. Regardless of the acquisition date, all anonymized T1 scans were pooled and the 23 vertebra units (VU: region between two virtual lines drawn through the middle of adjacent vertebral bodies) were extracted using an object detection system called SpineNet [16]. MR scans from F2305 were used for the training and validation, with respectively 7832 and 2205 VUs. The validation set and training set were split so that there was no overlap of patients in both sets. In addition, data from the AIN457A2209 (A2209) study acted as an independent test set [17]. The study included 27 patients with MRI scans, which translated into 1625 VUs that were used solely for testing purposes.

Each VU was pre-processed to have 9 slices (4 before and after from the central one) and labelled according to their anatomical regions (cervical, thoracic and lumbar). VUs from the sacrum were excluded from the pooled dataset. All VUs were rotated to align the VUs together (vertical line) and zoomed to have consistent size-ratio across VUs.

## Auxiliary classifier generative adversarial networks (ac-GAN)

The p-GAN model workflow employs auxiliary classifier generative adversarial networks (ac-GAN) [18] trained to generate synthetic VUs and associated VU regions. ac-GAN is a variation of generative adversarial networks [14], which utilizes a generator and discriminator to create realistic images. **Fig 2** shows the general structure of the ac-GAN model.

The training was done by iteratively improving the generator and discriminator. The loss consists of two components, the discriminative loss $L_D$ on distinguishing between real and fake images and an $\alpha$ weighted classification loss $L_C$ in the accuracy of predicting class labels, i.e.

$$L = L_D + \alpha L_C$$

where,

$$L_D = E[log\ P(real|X_{real})] + E[log\ P(fake|X_{fake})]$$

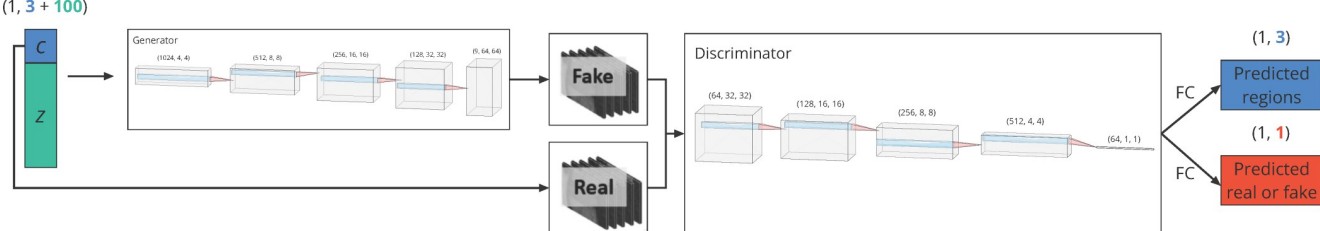

**Fig 2. The ac-GAN structure.** The generator (G) is a deep neural network that transforms a feature vector of the class label (c) and a random noise (z) to an image, and the discriminator (D) distinguishes between the real images and fake images and predicts the class labels for each image.

$$L_C = E[log\ P(\hat{c} = c|X)]$$

- **Pre-processing**: The same 9 slices of VUs were kept and rescaled from 112 x 224 to 64 x 64. For this specific dataset of VUs, although being 3D images, they were treated as 2-dimensional images with 9 channels and learnt with a 2D ac-GAN model, as the 3-dimensional structure in VUs were not very strong due to large slice thicknesses.

- **Feature vector**: Apart from the additional 3 dimensions for class labels (region encoded as a length 3 one-hot vector), a 100-dimensional standard Gaussian random noise was used as the input to the generator.

- **Generator**: The generator consisted of five 2-dimensional deconvolutions (transposed convolution) layers with 512, 256, 128, 64 and 9 output channels, respectively. A kernel of size 4 was used in all layers. The first layer had a stride of 1 and 0 padding to up sample the image size from 1 x 1 to 4 x 4, then the rest layers were set with stride size 2 and padding 1 to double the image size. Rectified Linear Unit (ReLU) activations were used in the first 4 layers and the hyperbolic tangent (tanh) activation was used in the last layer to restrict the output pixels within the range of [–1, 1].

- **Discriminator**: The discriminator consisted of five 2-dimensional convolution layers with 64, 128, 256, 512, 64 output channels, respectively, and each layer was set with kernel size 4. A stride size 2 and padding 1 was used in all layers but the last one, and the last layer was set with stride size 1 and padding 0. Leaky ReLU activations were used in all layers with a negative slope of 0.2. After passing through the convolution layers that extract information from images, VUs were then sent through two separate, fully connected layers with sigmoid and softmax activation, respectively, to produce the decision of real vs fake as well as the predicted classification label.

- **Loss:** The classification loss was weighted by $\alpha = 1$ during training.

- **Optimization:** Training was done with Adam optimizer18 with learning rate 1e-4 for both the generator and the discriminator.

- **Model Selection:** When performing hyperparameter search for training the model, visual inspection, adversarial and classification loss during training were monitored, as well as classification on the A2209 test set.

- **Synthetic Data:** To generate the synthetic dataset, VU location was first simulated from the empirical distribution of training data. The sampled VU location was then utilized to condition the generator and generate the entire image.

- **Reproducibility:** Trial data used for the GAN training is restricted, but synthetic VU data used for the diversity results are available. Additionally, all Python codes are fully available.

## Real vs. synthetic ResNet-18 performance

Prior to the training of *F_synthetic*, a synthetic dataset was created using the trained generator. The synthetic dataset size and class distribution was designed to approximate the scans from F2305 by generating 10,000 synthetic images, whereas the class assignment was sampled from a distribution $p_C, p_T, p_L$ = 0.25, 0.55, 0.2) where the probability was denoted for cervical, thoracic and lumbar, respectively.

Two ResNet-18 [19] based classifiers *F* were trained for the VU region. For the first classifier *F_real* F2305 was used as the training set, while the second classifier *F_synthetic* was trained solely on synthetic data. In both cases, A2209 was used as an independent test set. Both models used the same hyperparameters and an identical training procedure.

- **Optimization**: Stochastic gradient descent with a learning rate of $10^{-4}$ and momentum of 0.9 was used, and networks were trained for 20 epochs with a batch of 32.

- **Loss**: Cross-entropy loss was applied.

A single global ROC curve was computed for each classifier by averaging the class specific ROC curves using equal weights (macro-average). The 95 percentile confidence intervals for the AUCs were computed using bias-corrected and accelerated bootstrapping [20] with 2000 resamples.

## Image morphing

To generate the synthetic image, a tensor was inputted that concatenates a random noise $z \in R^{100}$ and a condition $c \in R^3$. The tensor *c* was a hot-encoding vector representing the vertebra location:

$$c_k = \begin{bmatrix} x_1 \\ x_2 \\ \dots \\ x_j \end{bmatrix} = \begin{cases} 1 & if\ j = k \\ 0 & otherwise \end{cases}$$

$$j, k \in \mathbb{N}^* \ and \ k \leq j$$

Thus, for all three locations:

$$c_{k=1\ (cervical)} = \begin{bmatrix} 1 \\ 0 \\ 0 \end{bmatrix},\ c_{k=2\ (thoracic)} = \begin{bmatrix} 0 \\ 1 \\ 0 \end{bmatrix},\ c_{k=3\ (lumbar)} = \begin{bmatrix} 0 \\ 0 \\ 1 \end{bmatrix}$$

To morph an image from a location to a new one, the tensor *c* can be modified to impact the generator, thus leaving untouched *z*. Linear interpolation was utilized between source condition $c_{source}$ and the target one $c_{target}$. The equation below describes condition $c_{k \to k'}^i$ for step *i* (out of *n*) to morph from *k* to *k'*:

$$c_{k \to k'}^i = \begin{bmatrix} x_1 \\ x_2 \\ \dots \\ x_j \end{bmatrix} = \frac{1}{n} \begin{cases} (n - i) & if\ j = k \\ i & if\ j = k' \\ 0 & otherwise \end{cases}$$

With,

$$k \neq k', i \in \{0, 1, 2, \ldots, n\} \ and \ n \in \mathbb{N}^*$$

Using the definition above, then to morph from cervical to lumbar the computation becomes:

$$c_{1 \to 3}^i = \frac{1}{n} \begin{bmatrix} (n-i) \\ 0 \\ i \end{bmatrix} \forall i \in \{0, 1, 2, \ldots, n\}$$

## Privacy

To evaluate privacy, a list of candidate samples was created that was composed of 1000 images: 1/3 from training, 1/3 from validation and 1/3 from test. For pairwise attacks, the comparison between synthetic and candidate was done in pixel space and embedding space. For the pixel space, similarity was computed as the minimum Euclidean distance between one candidate and all the synthetic samples. The embedding distance was obtained by training a Uniform Manifold Approximation and Projection (UMAP) trained on 5000 images (3000 from the trainset, 1000 from the validation set, and 1000 from the test set). The UMAP compressed the samples down to 64 features, and the feature distance was obtained as the minimum Euclidean distance on features. For the distribution attacks, the threshold for 2 points to be considered as neighbours was that their distance was part of the 1st percentile. A privacy leakage is defined as finding traces of the training dataset within the synthetic dataset.

## Results

To demonstrate the effectiveness of the proposed pGAN method, the algorithm was utilized to create a synthetic dataset of T1 magnetic resonance images of VUs, labelled with VU location regions such as cervical, thoracic and lumbar. The quality of the synthetic dataset was evaluated from the aforementioned three key metrics.

### Fidelity

The first assessment was to evaluate whether the synthetic VUs were visually indistinguishable from real VUs for a given location. **Fig 3A** shows a collection of real and synthetic data. Visually, synthetic samples look like real ones and expose different structures depending on their conditioned region location (i.e., cervical, thoracic and lumbar). Inspired from recent work on face morphing with GAN [21] **Fig 3B** shows that generator behaviour can be controlled by continuously changing the condition aspect from the input noise tensor [21]. This was achieved by using a linear regression between the source and target region wanted. This result demonstrated that the generator was correctly conditioned and could create specific VU locations on demand.

### Diversity

The real and synthetic datasets were evaluated to determine if they have similar global distribution while having distinct individual VU samples as shown in **Fig 4A**. To obtain a meaningful low dimensional representation, a UMAP transform was applied to the dataset of VUs. UMAP [22] is a powerful non-linear dimensionality reduction technique. It is similar under many aspects to t-SNE [23] but differs by the fact that the transformation from high dimension to low dimension is learnable. This was leveraged by training the UMAP only on real data and

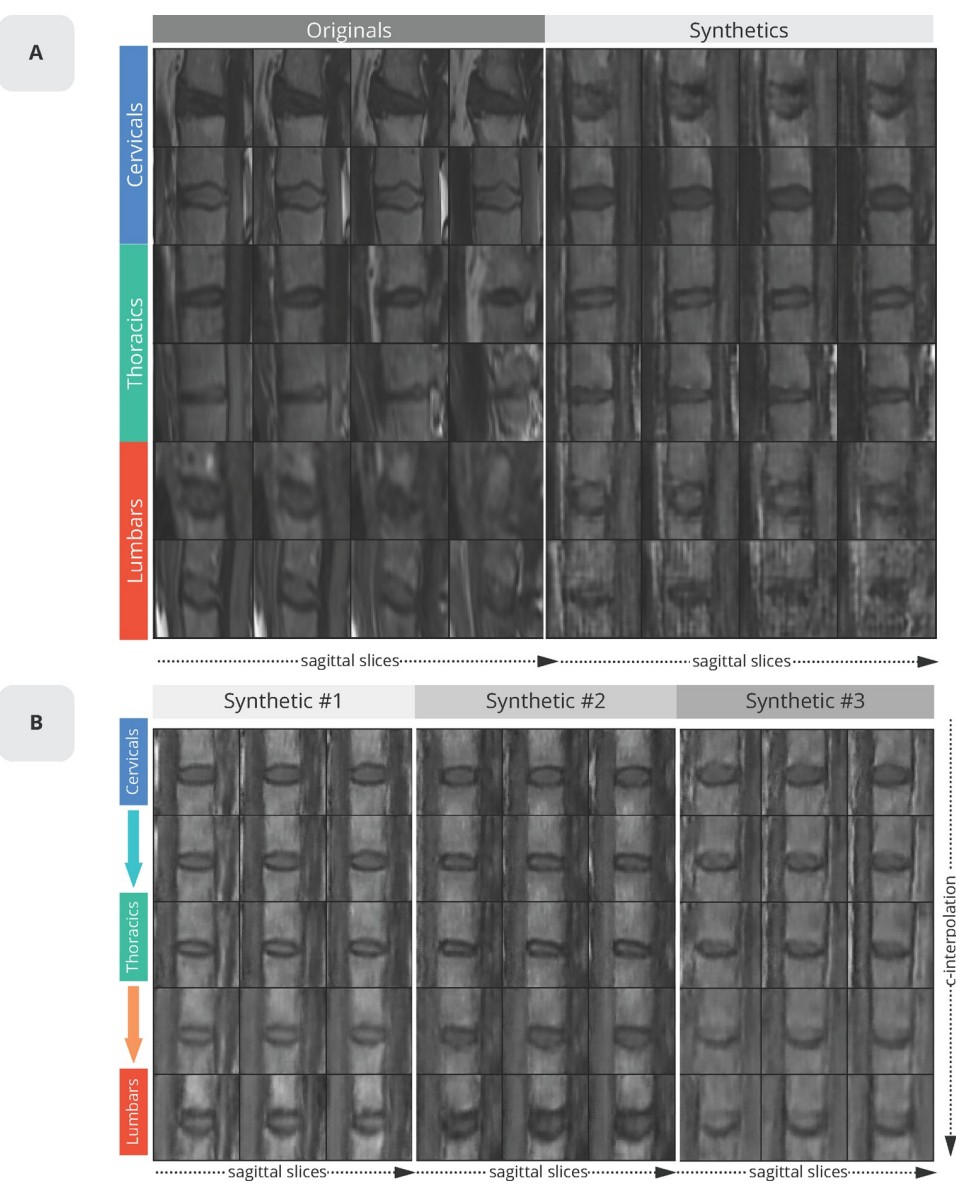

**Fig 3. Fidelity. A)** Examples of real and synthetic vertebrae and their corresponding vertebral unit (VU) locations. Each row shows 4 consecutive vertical slices for a real and fake VU (each slice is of size 64 x 64). The first two rows are cervical VUs, followed respectively by thoracic and lumbar. **B)** Morphing an image from one location to another was done by interpolating the condition *c* (keeping *z* untouched as the latent variable). One step for each morphing step was utilized, thus leaving intermediate states as 50% / 50% morph between two locations.

then applying the transform to unseen real data as well as synthetic data. The first finding was that VU location distribution was separable and similar between real and synthetic samples, showing that both datasets have similar global distribution. The second finding was that the synthetic VUs covered most of the distribution in the low dimensional representation space without overlapping with real VUs, implying that the synthetic samples were indeed distinct. Those findings confirmed that the GAN could synthetize data that behaves like real VUs and that its diversity (phenotypic differences) matched the real distribution as the mode was not observed to collapse. The quality of the synthetic dataset was further examined in terms of its capability to serve as a functional equivalent to a real dataset for downstream training tasks by training two

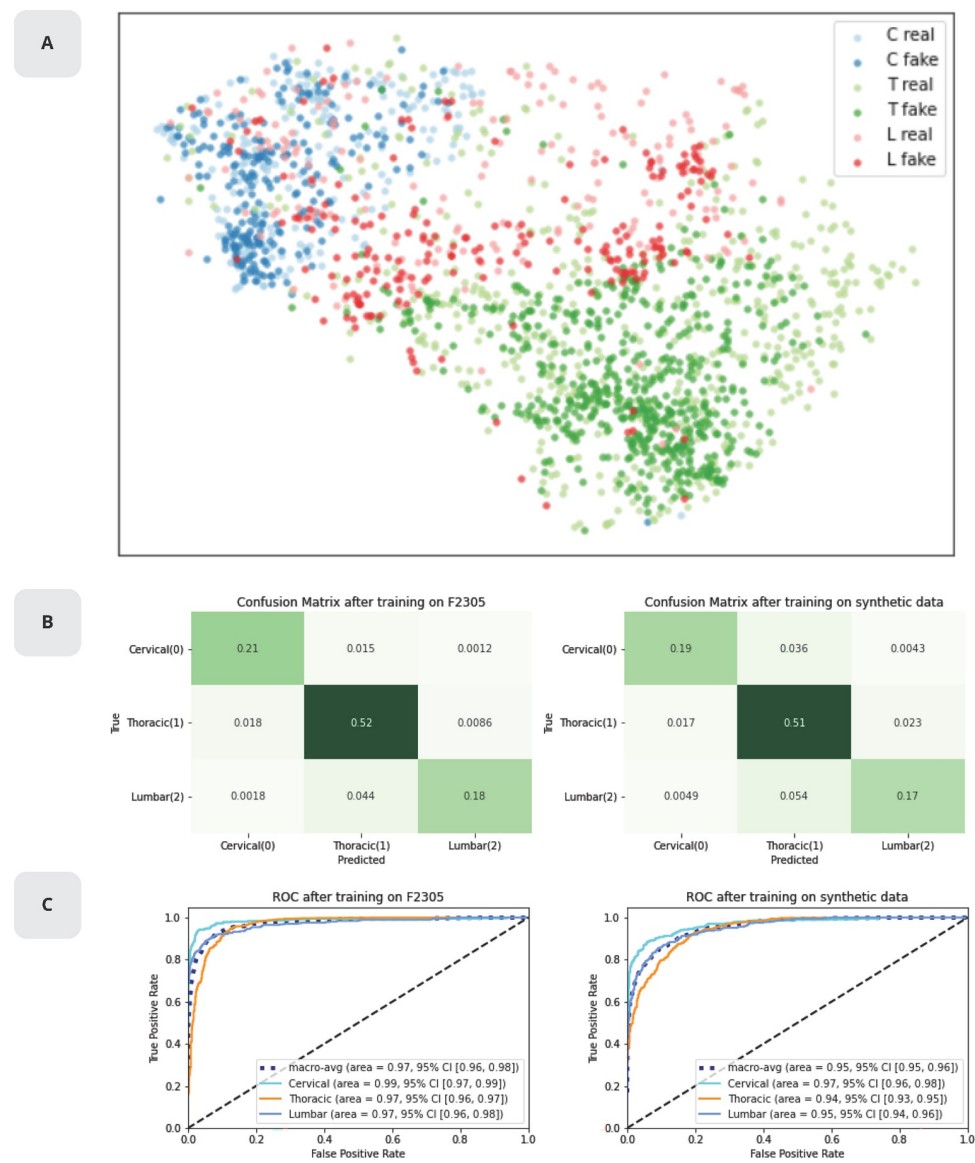

**Fig 4. Diversity. A)** Low dimensional UMAP representation of real and synthetic vertebral units with locations cervical, thoracic and lumbar labelled as blue, green and red respectively. **B)** Confusion matrices of $F\_real$ (left) and $F\_synthetic$ (right) when tested on A2209. The predicted class for a sample image $x_i$ corresponds to $max(F\_real(x_i))$ and $max(F\_synthetic(x_i))$ respectively. The class distribution of the test set corresponds to $p_C, p_T, p_L = (0.23, 0.55, 0.23)$. **C)** ROC and AUC for $F\_real$ (left) and $F\_synthetic$ (right) when tested on A2209. Every class specific ROC curve was computed by treating the other two classes as negative instances, effectively reformulating the classification for each class as a binary problem.

ResNet-18 [24] based classifiers $F\_real$ and $F\_synthetic$ with real and synthetic data respectively as seen in **Fig 4B**. **Fig 4C** shows the ROC curves for $F\_real$ and $F\_synthetic$. It was observed that, while $F\_synthetic$ slightly underperforms compared to $F\_real$, the synthetic dataset could approximate the distribution in the original data when tested for VU region. The slight discrepancy in performance was to be expected as the conservation of privacy could be expected to come at a cost in data quality. While, in principle, the performance of $F\_synthetic$ could asymptotically reach the one of $F\_real$, in practice there is a trade-off between overfitting the training set and replicating the classification statistics of a network trained solely on the training set.

## Privacy

In this section, the privacy leakage assumption was restricted to only sharing a synthetic and finite dataset rather than the trained generative model itself. This restriction made privacy attacks much more difficult because they could not be conducted by reverse engineering the network's output and there was no direct access to the generative model's latent space. This meant only attacks that rely directly on comparing candidate images with the synthetic dataset could jeopardize privacy. An obvious case of privacy threatening behaviour in GAN is overfitting. A major overfitting corresponds to a generator outputting synthetic images that are exact copies from training images. Other convergence issues could be problematic, such as cases where the generator only learns to reproduce a few samples from the training set (mode collapse).

Two main potential vulnerabilities were identified for a synthetic dataset:

- **Pairwise attacks**: where finding a synthetic image which is similar to a given sample would prove that this sample was used during training.

- **Distribution attacks**: where a high density of synthetic images cluster around one or a few real images.

To assess the robustness to these attacks, the synthetic images were simulated by using a "candidate dataset" (**S1 Fig**) and evaluated as to which ones could be traced back to the training set.

## Robustness to pairwise attacks

For robustness to pairwise attacks, the similarity between a candidate sample and all images were computed from the synthetic dataset. A relatively small L2 distance may indicate that the candidate was used for the generator training. The distribution of minimum distances between candidate and synthetic images are represented in **Fig 5A**. On this graph, the lowest distances are expected to be train-synthetic pairs and the largest ones to be test-synthetic due to some overfitting behaviour from the GANs. In the privacy-threatening scenario, images from the training set are easy to identify by a simple anomaly detection task. An anomaly detection with reference distances is available on **S2 Fig**. On the observed synthetic dataset however, it is impossible to reliably identify candidates originating from the training dataset as compared to the examples coming from validation set. This is also confirmed using an embedding space obtained by applying a UMAP. The synthetic dataset hence shows robustness to pairwise attacks, in both pixel and embedding space.

## Robustness to distribution attacks

To evaluate the robustness to distribution attacks, clusters of synthetic images were identified around a given candidate. The clusters were defined by counting the number of synthetic images in the candidate neighbourhood. These results are shown in **Fig 5B**. A large cluster was expected to form around candidates from the training set, with the other candidates to have a small number of neighbours. The privacy-threatening scenario (left figures) reflects this hypothesis. The observed behaviour of the synthetic data is shown on the two most right figures. Candidates from the test set rarely belong to large clusters unlike candidates from training or validation. In fact, the largest clusters overwhelmingly form around training samples. Even if this method can help identify a handful of training examples, it is ineffective for most of them. Indeed, the size of the training set and the synthetic dataset are around the same (around 10'000 samples). This means that on average, one training sample can be only traced

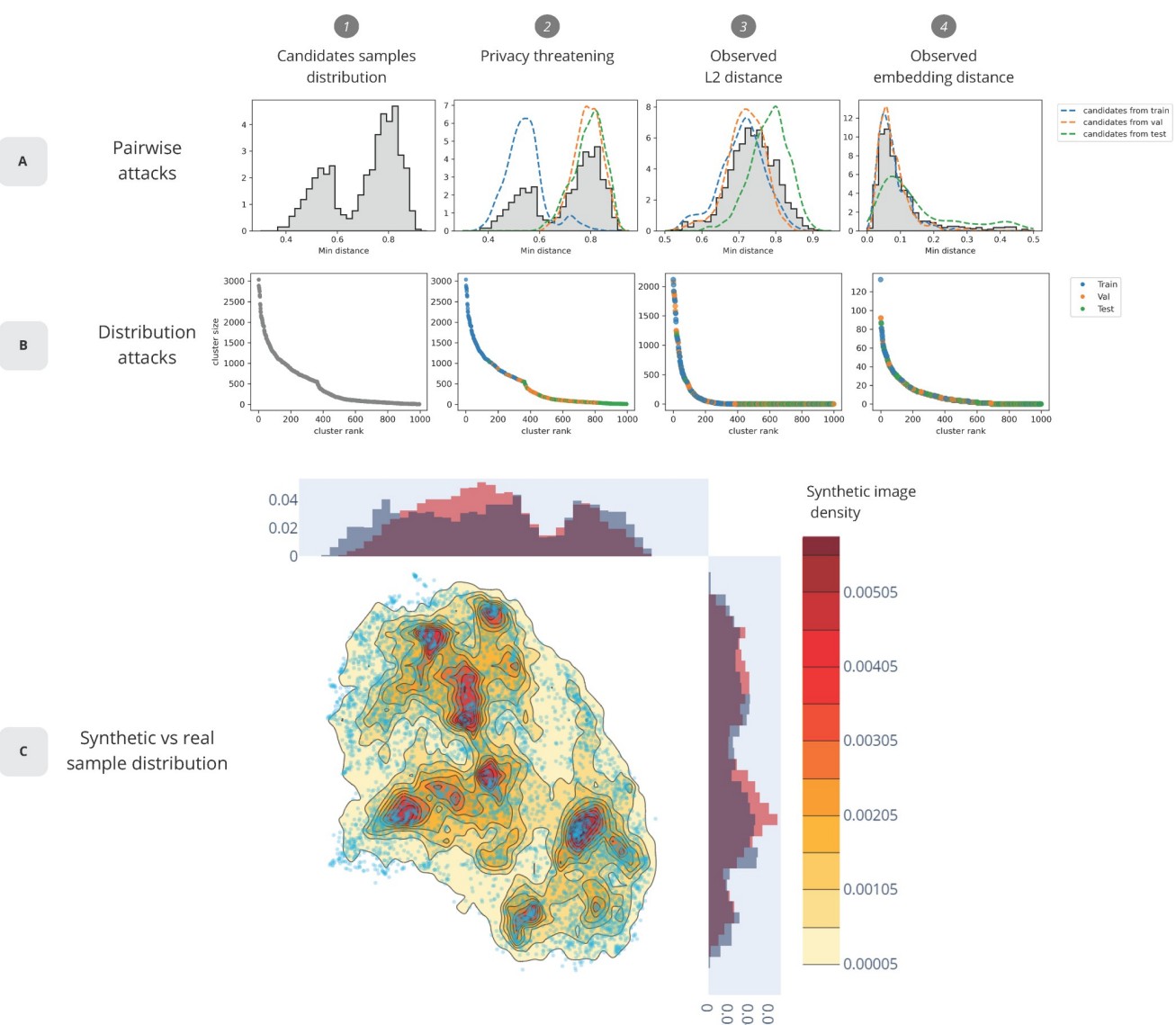

**Fig 5. Simulation of pairwise attacks and distribution attacks. A) Simulation of pairwise attacks.** A small distance indicates that the candidate is likely from the training set. Graphs 1–2 are toy examples of privacy threatening scenario with easily identifiable candidates from training. Graphs 3–4 are the observed behaviour of our synthetic dataset. **B) Simulation of distribution attacks**. A large cluster indicates that the candidate is likely coming from the training set. Graphs 1–2 are privacy threatening and graphs 3–4 are observed behaviour on the synthetic dataset. **C) 2D projection of the synthetic and training dataset.** Shades of yellow-red are density levels of synthetic images and blue dots are individual training examples.

back to one synthetic sample. This means that by construction, the generated synthetic dataset is safe from distribution attacks.

It is also worth noting that the embedding space was not uniformly populated. As seen on **Fig 5C,** a high density of synthetic images is in general linked to a high density of real images (blue dots on the figure) and not due to the GAN generation collapsing and only generating one type of sample. A detailed detection of training samples is available in the **S3, S4 Figs** and **S1 Table**.

## Discussion

Sharing data across research groups and institutions provides an opportunity to answer complex questions through the pooling of information and resources. However, there is a need to

share data safely and faster. Sharing a synthetic dataset overcomes the privacy and legal barriers to enable efficient collaborations. This method proposes to generate synthetic datasets of labelled medical images with a type of GAN model. The result satisfies the following qualities requisite of synthetic datasets:

- Fidelity: Realistic medical images and labels were produced. Generated images satisfied user defined conditions (spinal region)

- Diversity: The synthetic dataset preserved the analytical behaviour of the real data. Generated data had a similar distribution to real data in terms of variations.

- Privacy: When only the synthetic dataset was shared, real data were protected from identification. No real data could be identified from synthetic data only, nor limited access to real data (data leakage).

Data privacy protection in the age of data science and artificial intelligence has drawn a lot of attention, especially in the medical field [25]. One major way of compromising data privacy is membership inference attack [26], which aims to determine whether data from a target patient is included in the study. A general approach to protect data from such attacks is differential privacy [27], which masks real data with a carefully designed noise during training. It provides a strong protection of privacy and has been applied in a variety of problems, including synthetic data generation with GAN [27–30]. With differential privacy, privacy is protected at the cost of sample quality and diversity of the synthetic dataset. However, adding noise to the dataset is not a satisfactory solution to the problem, as it biases the synthetic distribution and, with images, noise only alters mathematical similarities while keeping general shapes and image content intact. This makes noise-based methods a particularly poor fit for image data privacy [26, 29]. Besides privacy protection, GAN has also been used in medical research, often as a data augmentation approach [31, 32], see for instance [29, 33–37]. For the presented analysis of the pGAN model workflow, a dataset of MRI spinal images with VU location labels (cervical, thoracic & lumbar) was used. A synthetic dataset was successfully created by using an ac-GAN to generate synthetic samples with their respective location labels. The synthetic dataset could safely be shared as a surrogate for the real dataset because it met consistency and privacy criteria. The synthetic samples were realistic and of high visual quality. The behaviour of the real dataset was replicated, and dataset-wide analysis, such as classification trained from scratch, performed to a similar level on both the synthetic and the real dataset. No sample from the synthetic dataset could be associated to a sample from the real dataset, making it impossible to trace back patient information from a synthetic sample.

This work provides a general tool for privacy protection when collaborating on medical datasets. For instance, the synthetic MRI spinal images could be used for developing and testing new prediction and classification models across institutions, for training readers / reviewers of MRI, etc. The results are not benchmarked against pure differential privacy (like additive Laplacian noise) because the effectiveness of such methods can be mitigated by convolution neural network (CNN) denoisers which would require an investigation on its own. Only privacy was evaluated through the re-identification of training samples (by outlier detection) which, in this case, is the only potential type of attack.

The results presented here show that the pGAN model does not replicate samples from the training set and tends to be robust to pairwise attacks. Some characteristics, the presence of artifacts for example, affect the similarity measures in pixel space, thus rendering the comparison of real and synthetic samples particularly difficult. Comparing in feature space does not solve the problem because the only embedding space where synthetic and training images share similar features is the pGAN embedding space itself. Density attacks seem to be more

dangerous than pairwise attacks because they do not require an exact match between a synthetic and a candidate sample. Simply having multiple synthetic images that are like a candidate image could already be enough to conclude on its origin. Successfully, the synthetic dataset is robust to density attacks. With approximately 1.2 synthetic data points per real data point, the synthetic dataset is too small to create meaningful clusters. This means clusters will either be large but not discriminative or too small to be relevant.

## Limitations

Ensuring privacy is a challenging aspect of this project, as defining a robust method to assess similarity between images is complex [38]. Due to sample size limitation, VUs were investigated instead of full spines images. This approach increased the effective sample size by pooling all time points and all VU locations, thus transforming the 108 patients into 7832 VUs. The GANs were able to be trained with sufficient data, at the cost of limiting the generation of synthetic images to the smaller VU piece. Conditioning the pGAN on a clinical metric was also investigated (VU inflammation score) but did not produce the desired results as the classifier task is more challenging than the prediction of a VU location. Several reasons can explain this: imbalance between the number of positive and negative samples, as well as the intra-reader discrepancies, led to similarly noisy results that proved too hard to analyse. Finally, in the present approach, the synthetic dataset has a fixed privacy tolerance that is a factor of the real dataset and the pGAN model convergence but cannot be modulated based on specific needs. Privacy was computed ad hoc rather than optimized, which may limit the control on the level of privacy needed for some applications.

## Future steps

Images generated by the pGAN model were realistic, informative and private. Additional analyses could be performed to further confirm those attributes. Privacy was computed as the distance of the synthetic images from the original datasets (in a form of attacks). The chance of generating a synthetic image with pGAN that resembles a real image is low, as the high dimensionality of the sampling space minimises the probability of synthesising an image with parameters identical to an original image. Further efficiencies in the method can be considered when confirming that the network did not reproduce a real sample, as compared to after each sample generation. A direction for further research to enhance the sample privacy could be to determine where the real images lie in the latent space of the generator and then create a careful sampling strategy to avoid sampling around these points. A future focus would be to generate complete synthetic spines instead of independent VUs. This would require balancing the trade-off between VU consistency and memory, since generating 21 VUs at once is extremely memory intensive and only a weak correlation would be required.

## Conclusion

A robust synthetic MRI dataset was generated from spinal MRI VU scans using an ac-GAN method. Analysis performed across three key metrics (image fidelity, sample diversity, and dataset privacy) demonstrated that the ac-GAN method produced synthetic images that were realistic samples with similar characteristics as the original dataset and yet could not be mistaken for the original images. As large imaging datasets are difficult to obtain for use in training, machine learning, and exploratory applications, especially when considering patient studies and privacy limitations, the possibility to generate synthetic images presents a valuable opportunity to pursue further.

## Supporting information

**S1 Fig. Candidate pair definition.** Column A illustrates an example of a pairwise attack of a synthetic dataset. The green digit from candidates is closely matching the green digit from the synthetic dataset meaning this candidate is likely in the trainset and privacy is not preserved. Column B illustrates an example of a distribution attack on a synthetic dataset. While not being an exact match, the green digit is very similar to the digits from synthetic. This indicates that the green digit is likely present in the training set.
(PDF)

**S2 Fig. Privacy attack scenarios.** In this experiment, the abnormally low distances between candidates and synthetic images can be identified by anomaly detection. An abnormal sample is any sample that diverges largely from the distance test-synthetic (green distribution). For the synthetic dataset, validation samples are also classified as outliers, privacy was preserved.
(PDF)

**S3 Fig. ROC curves.** ROC for classification of candidate samples from train vs validation (orange) and train vs test (green) are presented. A) Pairwise attacks. The classification is under the assumption candidates with the lowest distances are likely from training. B) Distribution attacks. The classification is done by associating large clusters to training samples. A high AUC means it is easy to classify training from other samples.
(PDF)

**S4 Fig. Cut-off curves.** The cut-off is defined as the threshold for a point to be considered an outlier. The orange curve shows the proportion of outliers from training vs outliers from validation. The green curve shows the proportion of outliers from training vs outliers from test. A privacy threatening case corresponds to almost all the outliers coming from the training set.
(PDF)

**S1 Table. Classification of candidate origin.** Proportion of origin for different candidates for a fixed cut-off. Cut-off 50 means the top 50 largest clusters or smallest distances are considered outliers (50 is top 5% of the 1000 candidate images). In extreme privacy threatening scenarios, it was expected that all training sample can be identified as outliers, meaning they would represent 100% of the top 333 candidates.
(PDF)

## Acknowledgments

The authors thank the study investigators for their contributions.

## Author Contributions

**Conceptualization:** Hanxi Sun, Jason Plawinski, Sajanth Subramaniam, Thibaud Coroller.

**Data curation:** Hanxi Sun, Jason Plawinski, Sajanth Subramaniam, Thibaud Coroller.

**Formal analysis:** Hanxi Sun, Jason Plawinski, Sajanth Subramaniam, Thibaud Coroller.

**Supervision:** Amir Jamaludin, Timor Kadir, Aimee Readie, Gregory Ligozio, David Ohlssen, Mark Baillie.

**Writing – original draft:** Hanxi Sun, Jason Plawinski, Sajanth Subramaniam, Thibaud Coroller.

**Writing – review & editing:** Amir Jamaludin, Timor Kadir, Aimee Readie, Gregory Ligozio, David Ohlssen, Mark Baillie.

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
