## [Decision Letter · Decision Letter 0]

3 Aug 2022

PONE-D-22-03138A deep learning approach to private data sharing of medical images using conditional generative adversarial networks (GANs)PLOS ONE

Dear Dr. Coroller,

Thank you for submitting your manuscript to PLOS ONE. After careful consideration, we feel that it has merit but does not fully meet PLOS ONE’s publication criteria as it currently stands. Therefore, we invite you to submit a revised version of the manuscript that addresses the points raised during the review process.

Dear authors

Based on the comments raised from the reviewer and my evaluation, I recommend a major revision of the manuscript.

The reviewer raised very important parameters in the evaluation of your method that should be addressed

before further recommendation of the manuscript.

We look forward to receiving your revised manuscript.

Kind regards,

Stavros I. Dimitriadis

Academic Editor

PLOS ONE

Journal Requirements:

2. Please provide in your manuscript text information on the location of the A2209 dataset, either as a literature reference or as a URL link to the location of the dataset.

4. Thank you for stating the following in the Competing Interests:

“I have read the journal's policy and HS, AJ, TK declared no competing interests.

I have read the journal's policy and TC, MB, JP, SS, GL, AR, DO are employees of

Novartis.”

We note that one or more of the authors have an affiliation to the commercial funders of this research study : Novartis

“The authors thank the study investigators for their contributions. The studies were funded by Novartis Pharma AG, Basel, Switzerland, in accordance with Good Publication Practice (GPP3) guidelines (http://www.ismpp.org/gpp3). “

“The study was sponsored by Novartis Pharma AG. Novartis personnel and academic

advisors from Oxford Big Data Institute (BDI) designed the project.”

7. Please amend your manuscript to include your abstract after the title page.

Additional Editor Comments (if provided):

Dear authors

Based on the comments raised from the reviewer and my evaluation, I recommend a major revision of the manuscript.

The reviewer raised very important parameters in the evaluation of your method that should be addressed

before further recommendation of the manuscript.

Reviewers' comments:

Reviewer's Responses to Questions

**Comments to the Author**

1. Is the manuscript technically sound, and do the data support the conclusions?

Reviewer #1: Yes

2. Has the statistical analysis been performed appropriately and rigorously? 

Reviewer #1: Yes

3. Have the authors made all data underlying the findings in their manuscript fully available?

Reviewer #1: No

4. Is the manuscript presented in an intelligible fashion and written in standard English?

Reviewer #1: Yes

5. Review Comments to the Author

Reviewer #1: Thank you for writing and submitting this manuscript entitled, ““GANs for developing synthetic MRIs.”

Sharing sensitive data under strict privacy regulations remains a crucial challenge in advancing medical research and especially deep learning systems that require large amounts of data to learn meaningful representations robustly. Recently, generative adversarial networks (GANs), have demonstrated the capability to generate realistic, high-resolution synthetic image datasets as a potentially viable approach to privacy-preserving data sharing.

In this study the authors trained an auxiliary classifier GAN to generate partial images of the spine, called vertebral units (VUs), to generate synthetic dataset and conduct an analysis on its three core properties: image fidelity, sample diversity and dataset privacy.

I think the study offers valuable practical models under which insights derived from synthetic images are similar to those that would have been derived from real data.

Several limitations/comments:

- Lack of large enough dataset preventing investigation of full spine images instead of vertebral units.

- Difficulty to use the pGAN on clinical metrics which is clinically important. It is also important to note that despite several tested scenarios the presented study does not provide any mathematical guarantees for the privacy of the synthetic data, and there are likely cases in which privacy would be breached in practice.

- Assessment of fidelity is not presented in a reader study, just by one figure.

-Why aren't the training and validation datasets from F2305 study and test dataset from A2209 study divided by 23 VUs of each patient? How were the numbers of VUs generated?

- How many patients were used for the test datasets?

- I guess Fig 2 in the text is fig 5 in the actual images, Fig 3/4/5 in the text are Fig 2/3/4? – please revise.

-I guess that cases with spinal hardware have been excluded?

6. PLOS authors have the option to publish the peer review history of their article (what does this mean?). If published, this will include your full peer review and any attached files.

Reviewer #1: No

---

## [Author Response · Author response to Decision Letter 0]

18 Oct 2022

Responses to editorial comments:

We thank the editor for the feedback. We have edited the manuscript accordingly.

1. We have made the necessary formatting changes as per the template provided.

2. We have added reference in the manuscript line 122 - Baeten et al. Anti-interleukin-17A monoclonal antibody secukinumab in treatment of ankylosing spondylitis: a randomised, double-blind, placebo-controlled trial. Lancet 2013

3. We have added “Novartis Pharma AG” as the funder in the appropriate section in the submission site

4. We have included the updated funding statement and competing interest statement in the revised cover letter

5. We have linked the corresponding author’s ORCID ID with the manuscript

6. We have removed the funding statement from Acknowledgement and included the funding statement and competing interest statement in the revised cover letter

7. Abstract has been included in the manuscript

8. Captions for Supporting Information has been updated and added at the end of the manuscript. 

Reviewer's Responses to Questions

We thank the reviewer for the constructive feedbacks. We edited the manuscript accordingly to them.

1. Lack of large enough dataset preventing investigation of full spine images instead of vertebral units.

We agree with the reviewer. The lack of data for full synthetic MRIs was the main reason why we focused on vertebras units (VUs) instead of whole images. By using VUs, we increased our effective sample size by pooling all time steps and VU locations together. We acknowledge this in the Limitations (line 421). Following-up on the comment, we added further comments on the issue (line 422). 

We acknowledged the data limitation to generate 21 vertebras units (VUs) at once (i.e., a whole spine) at line 447 in the Discussion. 

2. Difficulty to use the pGAN on clinical metrics which is clinically important. 

We added further detail on these results. To follow good science practices, we shared negative results from this experiment as well as successful one (i.e., VU location). We also added our interpretation of why the clinical conditioning failed as opposed to the location. We edited the manuscript accordingly in the Limitation (line 428) 

3. It is also important to note that despite several tested scenarios the presented study does not provide any mathematical guarantees for the privacy of the synthetic data, and there are likely cases in which privacy would be breached in practice.

We tested the most common scenario under clear condition. The manuscript does not make a claim for a mathematically guaranteed privacy model, but rather demonstrates how the model was tested under real scenarios (pairwise and distribution attacks).

4. Assessment of fidelity is not presented in a reader study, just by one figure.

We agree with the reviewer. The fidelity was indeed assessed by generating synthetic images. We generated many images and presented it internally. While this approach is by no means meant to quantify their fidelity, it gave us a high confidence on their visual quality. We then exhaustively tested our images to train a model from scratch (diversity) to ensure that they are representative and informative. 

5. Why aren't the training and validation datasets from F2305 study and test dataset from A2209 study divided by 23 VUs of each patient? How were the numbers of VUs generated?

We thank the reviewer for this question. The number of samples is counted as VUs and not as patient because VUs were treated as independent of their spinal location to increase our effective sample size. We edited the manuscript accordingly in the methods and discussion. 

6. How many patients were used for the test datasets?

The independent test set (A2209) included 27 patients. We edited the manuscript line 122 to reflect this information. We added the reference to the manuscript as well:

Baeten et al. Anti-interleukin-17A monoclonal antibody secukinumab in treatment of ankylosing spondylitis: a randomised, double-blind, placebo-controlled trial. Lancet 2013.

7. guess Fig 2 in the text is fig 5 in the actual images, Fig 3/4/5 in the text are Fig 2/3/4? – please revise.

We thank the reviewer for this comment and apologize with the mistake done during submission time. We changed the figure numbers and edited the text accordingly.

8. guess that cases with spinal hardware have been excluded?

During the collection of the MRI scans, patients with pacemakers, aneurysm clips, artificial heart valves, ear implants, metal fragments or foreign objects in the eyes, skin or body that rendered the patient unable to undergo MRI were excluded from submitting MRI scans, therefore patient scans utilized in this analysis are not likely to include spinal hardware.

---

## [Editor Report · Decision Letter 1]

27 Dec 2022

A deep learning approach to private data sharing of medical images using conditional generative adversarial networks (GANs)

PONE-D-22-03138R1

Dear Dr. Coroller,

We’re pleased to inform you that your manuscript has been judged scientifically suitable for publication and will be formally accepted for publication once it meets all outstanding technical requirements.

Kind regards,

Sathishkumar V E

Academic Editor

PLOS ONE

Additional Editor Comments (optional):

Reviewers' comments:

<quillbot-extension-portal></quillbot-extension-portal>

---

## [Editor Report · Acceptance letter]

26 Jan 2023

PONE-D-22-03138R1 

A deep learning approach to private data sharing of medical images using conditional generative adversarial networks (GANs) 

Dear Dr. Coroller:

I'm pleased to inform you that your manuscript has been deemed suitable for publication in PLOS ONE. Congratulations! Your manuscript is now with our production department. 

Kind regards, 

on behalf of

Dr. Sathishkumar V E 

Academic Editor

PLOS ONE